# PC-SwinMorph: Patch Representation for Unsupervised Medical Image Registration

## Abstract

Medical image registration is a critical task for several clinical procedures. Manual realisation of those tasks is time-consuming and the quality is highly dependent on the level of expertise of the physician. To mitigate that laborious task, automatic tools have been developed where the majority of solutions are supervised techniques. However, in the medical domain, the strong assumption of having a well-representative ground truth is far from being realistic. To overcome this challenge, unsupervised techniques have been investigated. However, they are still limited in performance and they fail to produce plausible results. In this work, we propose a novel unsupervised framework for image registration that we called PC-SwinMorph. The core of our framework is two patch-based strategies, where we demonstrate that patch representation is key for performance gain. We first introduce a patch-based contrastive strategy that enforces locality conditions and richer feature representation. We also introduce a novel patch stitching strategy based on a 3D window/shifted-window multi-head self-attention module to eliminate artifacts from the patch splitting. We demonstrate, through a set of numerical and visual results, that our technique outperforms current state-of-the-art unsupervised techniques.

## 1 Introduction

Image registration is a fundamental task in medical image analysis, which aims at finding a mapping that aligns an unaligned image to a reference one. The estimated spatial mapping (deformation field) seeks to best align the anatomical structure of interest. These Techniques are relevant for several tasks in clinical practice including image-guided surgery (Aviles et al., 2016; Han et al., 2021b), segmentation (Fu et al., 2017; Liu et al., 2019) and image reconstruction (Lee & Kang, 2003; Liu et al., 2021a). Mathematically, registration involves two key images: the unaligned image, denoted as $x$, and the reference image, represented as $y$. The primary objective is to find an optimal spatial mapping (deformation field), denoted as $z$. This mapping includes a spatial transformation function $\Psi_z$. By applying $\Psi_z$ to $x$, we can effectively align the coordinates of $x$ with those of the reference image $y$. Consequently, the transformed image, expressed as $x \circ \Psi_z$, achieves alignment with the reference image $y$. The optimization problem can be written as:

$$\hat{\Psi}_z = \arg\min_z \mathcal{L}_{recon}(y, x \circ \Psi_z) + \lambda\mathcal{L}_{smooth}(z)$$

where $\mathcal{L}_{recon}$ is a reconstruction loss that measures image similarity between the two input images, and $\mathcal{L}_{smooth}$ is a regularzation term to make $z$ is smooth enough, and $\lambda$ is the hyperparameter.

The outcome of those tasks greatly depends on the quality and efficiency of the registration technique. Although traditional image registration techniques (Rueckert et al., 1999; Vercauteren et al., 2009; Beg et al., 2005; Hart et al., 2009) are able to generate a good mapping between images, they build upon costly optimisation schemes, which limits their efficiency when using a large volume of data. With that limitation in mind, several deep learning techniques have been proposed for registration.

A major category of approaches is supervised image registration techniques (Yang et al., 2017; Sokooti et al., 2017; Rohé et al., 2017; Cao et al., 2017; 2018), where a good quality ground-truth is required for training.

However and unlike other tasks in image analysis, it is very difficult to obtain high-quality ground-truth deformation fields or segmentation masks. Although a good mapping can be obtained from traditional methods or using synthetic data, this drawback hinders the performance and feasibility of those techniques in clinical practice.

To mitigate the aforementioned strong requirement of supervised methods, a body of literature has been devoted to developing unsupervised techniques (Mok & Chung, 2020; Liu et al., 2020; Kim et al., 2021; Ye et al., 2021; Liu et al., 2022). Those techniques have proposed different network mechanisms and explicit regularisers embedded in the architectures to enforce better correspondences between images. However, unsupervised techniques are still limited in performance compared to supervised methods. This is due to the lack of high-quality ground-truth that introduces challenges such as failure in long-range correspondences. With the aim to alleviate this problem, recent techniques yet scarce have used vision transformers (ViT), where the self-attention mechanism (Chen et al., 2021b; Zhang et al., 2021) is key for improving the correspondence of the image. Another strategy reported for unsupervised image registration is the use of contrastive mechanisms (Liu et al., 2020) for improving feature representation, and therefore, enforcing a better mapping estimation between images. Although these techniques have reported improved performance for unsupervised image registration, it is still limited.

In this work, we proposed a framework for unsupervised image registration, which we call PC-SwinMorph (**P**atch **C**ontrastive Strategy with **S**hifted-**win**dow multi-head self-attention). For a fair comparison to the state-of-the-art techniques, we use as backbone Voxel**Morph**. Medical images are more complex than natural images due to the anatomical structures such as the curved and convoluted patterns in brain scans, where fine details are of clinical relevance. *We then hypothesise that patch embeddings are a more meaningful representation for performance gain in medical data. This is due to the spatial structure of the patch that allows capturing not only global but, more importantly, also local anatomical representations. Our PC-SwinMorph then enforces more meaningful feature representation whilst enforcing local and global structure representation.* Our contributions are as follows:

- We propose a patch-based framework for unsupervised image registration, in which we highlight a patch-based contrastive strategy for enforcing a better fine detailed alignment and richer feature representation.
- We introduce a novel patch stitching strategy to alleviate the splitting effect caused by the patch representation. To do this, we use the 3D window/shifted-window multi-head self-attention module (3D W-MSA and 3D SW-MSA) to enable information exchange between different patches.
- We evaluate our framework using two major medical benchmark datasets CANDI and LPBA40. We demonstrate from the numerical and visual results that our two patch-based strategies lead to better performance than the state-of-the-art techniques for unsupervised registration.

## 2 Related Work

The problem of image registration has been extensively investigated in the literature (Fu et al., 2020; Zou et al., 2022), in which solutions broadly divide into classic techniques e.g. (Rueckert et al., 1999; Vercauteren et al., 2009; Beg et al., 2005; Hart et al., 2009) and learning-based methods e.g. (Yang et al., 2017; Shen et al., 2019; Kim et al., 2021; Liu et al., 2021a). Although, classic techniques have demonstrated potential results, a major bottleneck is the costly optimisation schemes needed for obtaining plausible results. The second category is the focus of our interest in this work. In this section, we review the existing techniques.

### 2.1 Learning-based Techniques for Image Registration

A set of techniques have been proposed for supervised image registration, where convolutional neural networks (CNNs) are a de facto standard in the models; for example, the works of that (Yang et al., 2017; Sokooti et al., 2017; Rohé et al., 2017; Cao et al., 2017; Liu et al., 2021a; Cao et al., 2018). Whilst supervised techniques have reported great performance, they require the ground truth deformation fields or segmentation masks. This requirement is particularly difficult in medical image registration. Existing techniques mitigate somehow that constraint by either relying on using classic techniques for getting a good ground truth estimation or

using synthetic data. However, the registration performance is conditioned to the quality of the ground truth and/or the synthetic data pre-processing.

To overcome the practical limitations of supervised techniques, another set of solutions has focused on designing unsupervised models. The authors of (Krebs et al., 2018) use a statistical regularisation term to learn a low-dimensional stochastic parametrisation of the deformations. The works of that (de Vos et al., 2017; 2019) use B-spline parametrisation in a multi-stage framework, this technique enforces a coarse-to-fine learning process. Balakrishnan et al. (Balakrishnan et al., 2018; 2019) introduced VoxelMorph, a cross-correlation CNN unsupervised framework that includes a spatial transform layer. A deep recursive cascade architecture was introduced in (Zhao et al., 2019), where a core of the model is a shared-weight cascading strategy.

In more recent works, Mok et al. (Mok & Chung, 2020) introduced a symmetric diffeomorphic framework called SYMNet, where authors guarantee topology preservation by introducing an orientation-consistent regulariser. The authors of (Liu et al., 2020) proposed a contrastive registration architecture that fusions image-level registration and feature-level contrastive representation. CycleMorph was proposed in (Kim et al., 2021), which uses cycle consistency. A key point of that work is that cycle consistency can provide a form of implicit regularisation for topology preservation. The authors of that (Ye et al., 2021) introduced a bidirectional diffeomorphic network, that technique enforces topology-preservation and inevitability of the deformation.

**Connection with Image Segmentation.** There is an inherent connection between image registration and segmentation. One can use the computed mapping, between the unaligned and reference images, to project the segmentation mask of the reference image into the coordinate system of the unaligned image (Liu et al., 2020; Wang et al., 2020). This process is called registration-based segmentation (aka atlas-based registration). In this work, we use this observation to unify the registration and segmentation process within one framework, as the unification can be advantageous as it allows for simultaneous refinement of both tasks, potentially leading to more accurate and cohesive results.

## 2.2 Vision Transformers & Constrastive Learning

A great focus of attention has been given to Vision Transformer (ViT) (Dosovitskiy et al., 2020) due to their performance-speed gain in classification tasks. After the work of Dosovitskiy et al. (Dosovitskiy et al., 2020), several improvements have been proposed such as the techniques of (Chu et al., 2021; Han et al., 2021a; Wang et al., 2021) and for other tasks such as semantic segmentation and image detection (Liu et al., 2021b; Wang et al., 2021). However, there are only a few works that tackle the challenges of dealing with medical imaging tasks, where the focus has been mainly on segmentation e.g.(Chen et al., 2021a; Hatamizadeh et al., 2021).

By contrast, the works reported for medical image registration using transformers are scarce. The work in (Chen et al., 2021b) proposed a hybrid ConvNet-Transformer architecture for self-supervised volumetric image registration. The authors of (Zhang et al., 2021) proposed a dual transformer network, where a self-attention scheme considers the inter- and intra- image context. These initial works showed the potential of vision transformers for image registration. In particular, the advantage is integrating more easily not only local but also global embeddings. However, there are still limitations on how self-attention schemes can better and more efficiently handle the correspondences between images.

Let us also mention the closely connected problem to our technique – representation learning from the unsupervised contrastive learning perspective (Hadsell et al., 2006). Contrastive learning techniques seek to learn similarities between sample pairs without supervision. Following this perspective, several techniques have been proposed including that of (Chen et al., 2020; He et al., 2020; Tian et al., 2020). Whilst the majority of existing techniques have been mainly applied for classification and segmentation tasks (Chen et al., 2020; He et al., 2020; Zhao et al., 2021), the number of works reported for medical image registration is very limited. To the best of our knowledge, the work of Liu et al. (Liu et al., 2020) is the only one reported for unsupervised registration and the closest to our purpose. In that work, the authors embedded a contrastive feature representation in the registration network to enforce feature maps with richer information.

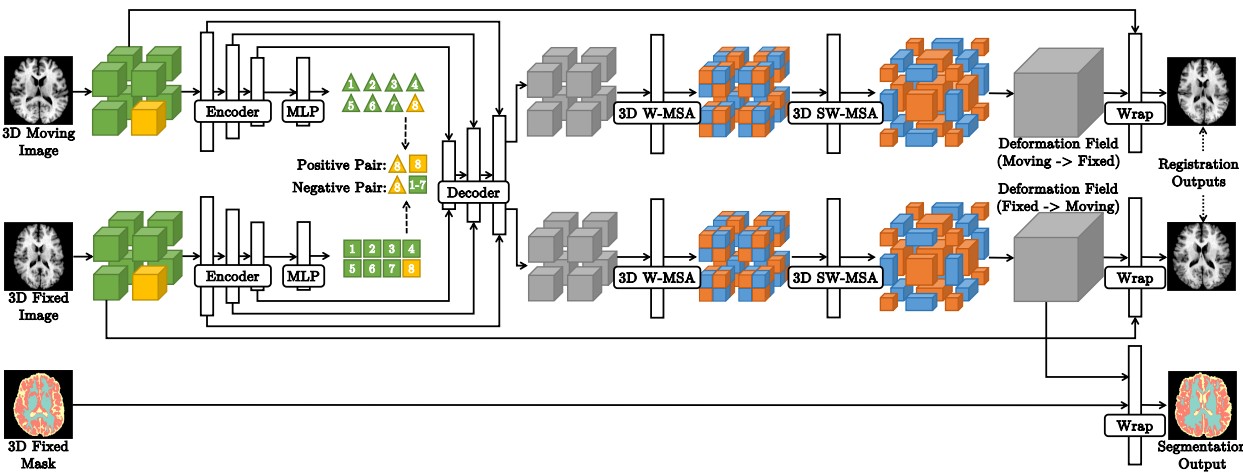

Figure 1: From left to right. Our PC-SwinMorph first generates non-overlap patches from the two input images, which are fed into two weight-shared CNN encoders. Followed by two MLP layers, the generated CNN features are projected to a latent space to obtain the patch representations (1-8 triangles and squares). Detailed patchwise contrastive mechanism is shown in Fig. 2. Based on patch representations, positive and negative pairs are sampled for patch-level contrastive learning (omitted from the figure). Then it recursively concatenates and enlarges the contrasted CNN feature with skip connection to reconstruct two sets of the deformation field patches. We use a 3D W-MSA and a 3D SW-MSA module to refine and stitch the deformation field patches to obtain the full deformation field. Using the full deformation fields, we warp the moving image to the fixed image, and vice versa. After the training registration process, we also adopt the full deformation field to transfer the segmentation mask for fixed masks to obtain the segmentation mask of the moving image. All inputs and outputs are 3D volumes, and all the operations are implemented in 3D.

## 3 PC-SwinMorph: A Patch Based Unsupervised Registration Framework

In this section, we first introduce the necessary preliminaries for our technique. We then describe our proposed unsupervised framework, called PC-SwinMorph for registration. We highlight two core strategies in our approach: i) patchwise contrastive learning, and ii) patches stitching using a shifted-window multi-head self-attention module.

### 3.1 Preliminaries & Workflow Overview

We first provide the essential preliminaries for our proposed framework. Let $x$ and $y$ denote the moving (unaligned) and fixed (reference) 3D images respectively, where $x, y \in \mathbb{Z}^{w \times h \times d}$. We refer to $w$, $h$, $d$ as the width, height, depth of the 3D images, where $\mathbb{Z}^{w \times h \times d} \subset \mathbb{Z}^3$. We also denote the deformation field from $x$ to $y$ as $z_{x \to y}$, where $z_{x \to y} \in \mathbb{Z}^{w \times h \times d \times 3}$. The deformation fields for 3D images are in a 4 dimensional space, i.e., $\mathbb{Z}^{w \times h \times d \times 3} \subset \mathbb{Z}^4$. The four dimensions refer to each channel containing the pixel moving information in the $w$, $h$, $d$ axis, respectively. Moveover, the deformation field $z_{x \to y}$ is parametrised with a spatial transformation function denoted as $\psi_{z_{x \to y}}$, such that, the registered results $x \circ \psi_{z_{x \to y}}$ is aligned with fixed image $y$.

The workflow of our PC-SwinMorph framework is displayed in Figure 1, which unifies unsupervised registration and segmentation tasks. Our framework uses an encoder to extract the CNN feature maps from the two given 3D images $x$ and $y$. We then seek to estimate two deformation fields $z_{x \to y}$ and $z_{y \to x}$ from the extracted CNN feature maps with skip connections. After we obtain the deformation fields, we perform registration by using a spatial transformer(Jaderberg et al., 2016) to warp the moving image $x$ and the deformation field $z_{x \to y}$. We can then obtain the registration output $(x \circ \psi_{z_{x \to y}})$, where $\circ$ denotes the warp operation. Similarly, we can also use a spatial transformer to warp the fixed image $y$ and the deformation field $z_{y \to x}$ to obtain $y \circ \psi_{z_{y \to x}}$.

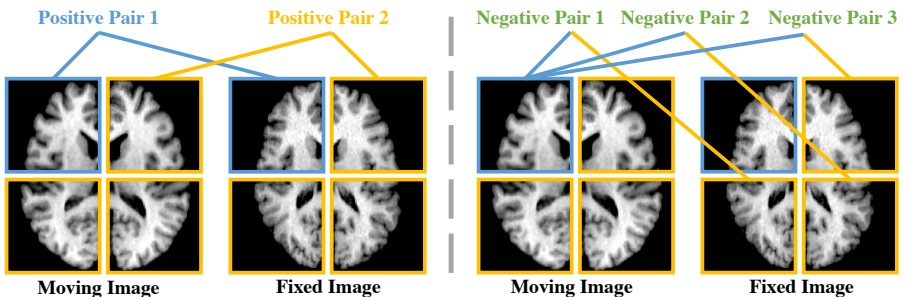

Figure 2: Patchwise contrastive mechanism. (Left side) Positive pairs are sampled from same-position patches in difference images, for example, the top-left patches of the moving and fixed image are **positive pair - 1**, and the top-right patches of the moving and fixed image are **positive pair - 2**. (Right side) Any pairs from difference position are sampled as negative pairs, such as the the top-left moving patch and the bottom-left fixed patch are **negative pair - 1**, and top-left moving patch and the top-right fixed patch are **negative pair - 2**. For visualisation purposes, the 3D volumes are represented as 2D slices.

After registration, we also use the spatial transformer to warp deformation field $z_{y \to z}$ and the segmentation mask of the image $y$. By doing this, we can obtain the segmentation mask for any image $x$. We underline that there is no segmentation mask used in the training registration process, and the segmentation mask is only used in the testing segmentation stage. Therefore, our framework is a unified unsupervised registration and segmentation network.

### 3.2 Patch-based Strategy I: Patchwise Contrastive Registration

A core of our technique is a patchwise strategy for image registration. It enforces an efficient and accurate registration output. We provide details next.

**Patch Generation.** We first generate patches from the two 3D images, instead of feeding directly the moving image $x$ and the fixed image $y$ as input. To do this, we evenly partition the 3D images into $n^3$ patches by dividing the image equally along the $x$, $y$, and $z$ axes without overlapping. This results in $n^3$ smaller cubic patches. We then assign a unique linear index to each patch, starting from 1 up to $n^3$. We denote the generated patches from the moving and fixed images as $p_i^x$ and $p_i^y$, where $i \in [1, 2, 3, ..., n^3]$ and $p_i^x$ and $p_i^y \in \mathbb{Z}^{\frac{w}{n} \times \frac{h}{n} \times \frac{d}{n}}$. For example, as shown in the Figure 1, we set $n$ as 2. Therefore, both moving images and fixed images are split into $2^3$ patches (8 patches).

**Patchwise Registration.** We first select a patch pair from the moving and fixed partitioned images at the same position ($p_i^x$ and $p_i^y$). We then feed them into a two-symmetric weight-shared 3D encoder to extract CNN feature maps (see the two encoders in Figure 1). We use a single decoder to integrate all the CNN feature maps generated by the two encoders. Specifically, we recursively combine the CNN feature maps from high-to-low level (low-to-high image resolutions) to reconstruct two deformation field patches ($p_i^{z_{x \to y}}$ and $p_i^{z_{y \to x}}$) that have the same resolution as the input patches ($p_i^x$ and $p_i^y$). See the decoder part in Figure 1. We repeat this process until every same-position patch pair has been fed through the encoder-decoder architecture to produce their corresponding two deformation field patches. The patch-wise deformation fields are reassembled into a comprehensive deformation map. This reassembled map is then used to warp the entire image, ensuring that each localised patch deformation contributes to the overall continuous transformation of the image.

After obtaining the patchwise deformation fields, we stitch them to produce the full deformation field ($\hat{z}_{x \to y}$ and $\hat{z}_{y \to x}$). To achieve this, we use a 3D Swin Transformer Block (Liu et al., 2021b). See the 3D W-MSA and 3D SW-MSA modules in Figure 1. The patch-wise deformation fields are reassembled into a comprehensive deformation map using the 3D Swin Transformer Block. This reassembled map is then used to warp the entire image, ensuring that each localised patch deformation contributes to the overall continuous transformation

of the image. We then use a spatial transformer to warp the moving image $x$ and the stitched deformation field $\hat{z}_{x \to y}$ to obtain the composition $x \circ \psi_{\hat{z}_{x \to y}}$, and do the inverse registration from $y$ to $x$.

As part of our unified framework for unsupervised registration and segmentation, we can now use the estimated deformation field to do segmentation tasks. More precisely, in the testing stage, we use a spatial transformer, to warp the segmentation mask of the fixed image $y_{seg}$ and (stitched) deformation field $\hat{z}_{y \to x}$, to generate the segmentation result $(y_{seg} \circ \psi_{\hat{z}_{y \to x}})$.

**Reconstruction & Regularisation Terms.** During the training process, we use two losses to guarantee a robust registration process. We first use a reconstruction loss, which enforces a plausible mapping to get the registered results as closest as possible to the template images. In our work, we use a normalised local cross-correlation loss (NCC loss) as the reconstruction loss. We denote the two input images as $x$ and $y$. Then the local mean of $x$ and $y$ at pixel $p$ are denoted as $\bar{x}(p)$ and $\bar{y}(p)$, respectively. The NCC loss is given as follows:

$$\mathcal{L}_{ncc}(x,y) = \sum_{p \in \Omega} \frac{\sum_{p_i}(x(p_i) - \bar{x}(p)) \cdot (y(p_i) - \bar{y}(p))}{\sqrt{\sum_{p_i}(x(p_i) - \bar{x}(p))^2 \cdot \sum_{p_i}(y(p_i) - \bar{y}(p))^2}}, \tag{1}$$

where the local mean $\bar{x}(p)$ and $\bar{y}(p)$ are calculated over a local window centered at pixel $p$ with window length of 9, and in the domain $\Omega \subset \mathbb{Z}^{w \times h \times d}$. The NCC loss is robust to changes in pixel magnitudes, such as differences in brightness or contrast between the images. This characteristic makes NCC particularly suitable for medical image registration, where images may have differing illumination or exposure settings. Our reconstruction loss reads:

$$\mathcal{L}_{recon} = \mathcal{L}_{ncc}(x \circ \psi_{\hat{z}_{x \to y}}, y) + \mathcal{L}_{ncc}(y \circ \psi_{\hat{z}_{y \to x}}, x). \tag{2}$$

We also include an L2 diffusion regulariser on the spatial gradients to obtain a smooth deformation field. It reads:

$$\mathcal{L}_{smooth} = \sum_{p \in \Omega} \|\nabla \psi_{\hat{z}_{y \to x}}\|^2 + \sum_{p \in \Omega} \|\nabla \psi_{\hat{z}_{y \to x}}\|^2. \tag{3}$$

**Patchwise Contrastive Loss.** In representation learning, contrastive learning has been a successful perspective to learn distinctiveness. The main idea of contrastive learning is to maximise the similarity between images and their augmented views, whilst minimising the similarity between images from different groups. In contrast to existing contrastive methods, which select the positive and negative pairs between images in the dataset, we proposed to select the positive and negative pairs within the image internally; as shown in Fig. 2. More precisely, after we recursively fed the moving and fixed patches $p_i^x$ and $p_i^y$ into the CNN encoders, we can obtain a set of high-level semantic CNN features maps for each patch. We use two linear projection layers (see the MLP in Figure 1) to map the high-level CNN semantic feature maps, for the moving and fixed patches, to a latent space, separately. Hence, the projected features are a calculated representation of the moving patch and fixed patch. We denote the projected features as $f_i^x$ and $f_i^y$ for the moving and fixed patch respectively, see the triangles and squares tagged as 1 to 8 in Figure 1. Between the two sets of projected features, we consider as a positive pair any part from the same partition position ($f_i^x$ and $f_i^y$), otherwise a negative pair ($f_i^x$ and $f_j^y$, where $i \neq j$). We then consider the following patch-wise contrastive loss (Park et al., 2020) from $f_i^x$ to $f_i^y$ expressed as:

$$\mathcal{L}_{contrast}^i(f_i^x, f_i^y) = -\log \frac{\exp(sim(f_i^x, f_i^y)/\tau)}{\sum_{j=1}^n \exp(sim(f_i^x, f_j^y)/\tau)}, \tag{4}$$

where $sim(u,v) = \frac{u^{\mathrm{T}} v}{\|u\| \cdot \|v\|}$ is the cosine similarity between $u$ and $v$. Moreover, $\tau$ is a temperature hyperparameter set as 1, and $n$ is the number of patches. Similarly, the the contrastive loss from $f_i^y$ to $f_i^x$ which reads:

$$\mathcal{L}_{contrast}^i(f_i^y, f_i^x) = -\log \frac{\exp(sim(f_i^y, f_i^x)/\tau)}{\sum_{j=1}^n \exp(sim(f_i^y, f_j^x)/\tau)}. \tag{5}$$

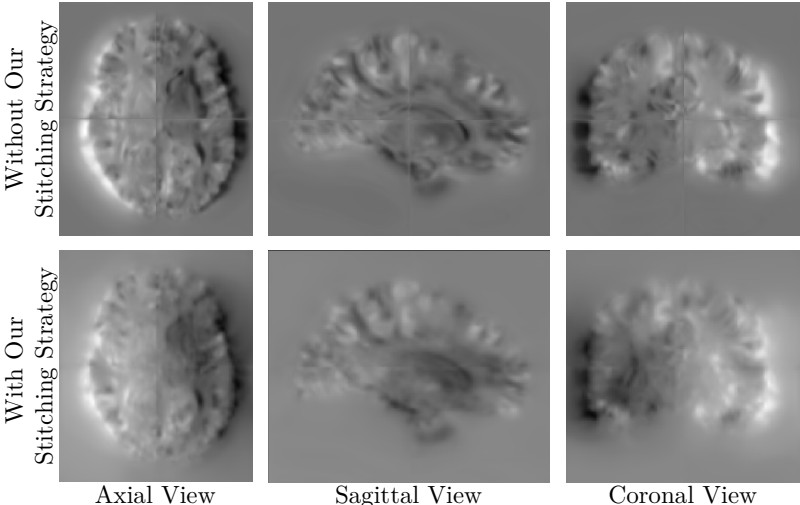

Figure 3: Deformation field learned with and without the proposed stitching strategy. Three columns show the sample slice of the 3D deformation field from different views. Direct patch stitching (first row) yields results with clear discontinues between patches, whilst stitching with the proposed multi-head self-attention (second row) produces smooth and continuous deformation fields with the gap between patches nearly invisible.

Hence, the final patchwise contrastive loss is calculated as a weighted sum of equation 4 and equation 5 as:

$$\mathcal{L}_{contrast} = -\frac{1}{2n}\sum_{i=1}^{n}(\mathcal{L}_{contrast}^i(f_i^x, f_i^y) + \mathcal{L}_{contrast}^i(f_i^y, f_i^x)). \tag{6}$$

### 3.3 Patch-based Strategy II: Patches Stitching with Shifted-window Multi-head Self-attention

The multi-head self-attention (MSA) module (Vaswani et al., 2017) has been proved to be an effective tool for capturing content relations from within the image. Unlike CNN-based networks that use a hierarchical convolutional layer to expand the reception field from local to global, the MSA module can directly calculate the similarity of all non-overlapping patches within the image. In our work, to alleviate the blurring effect from the process of direct patch stitching (see the top row of Figure 3), we propose to stitch the patches with multi-head self-attention. We utilise the multi-head self-attention mechanism, which is adept at capturing long-range dependencies within the data. This characteristic is particularly beneficial for smoothing the stitched image, as it allows the model to consider global information and achieve coherent integration of patches. Specifically and for computational efficiency, we use the improved 3D window/shifted-window multi-head self-attention (3D W-MSA and 3D SW-MSA) from Swin Transformer (Liu et al., 2021b).

**3D W-MSA.** To align with the definition of Swin Transformer, we define each deformation field patch $p_i^{z_{x \to y}}$ as a window. Each window is further splited evenly into $m \times m \times m$ small non-overlapping regions $r_{ij}^{z_{x \to y}}$ where $i \in n^3, i \in m^3$. The original MSA performs the computation directly on the regions of an image with a size of $w \times h \times d$. The W-MSA calculates region relations inside windows, significantly reducing computation time. The computational complexity of the two modules are listed as follow:

$$\Omega(MSA) = 4 \cdot whd \cdot C^2 + 2 \cdot (whd)^2 \cdot C$$
$$\Omega(W-MSA) = 4 \cdot whd \cdot C^2 + 2 \cdot m^3 \cdot (whd) \cdot C, \tag{7}$$

where the time complexity for the MSA module is quadratic to volume size $whd$, the time complexity for the W-MSA module is linear to volume size where $m$ is set as 4, and C is the image spatial channel number 3 in our experiment. Hence, with the W-WSA module, the computational time is fast, especially apply to 3D medical images.

**3D SW-MSA.** *One core disadvantage of 3D W-MSA is that it lacks information exchange between windows*, since all the computation is performed on regions within a window, which means that simply applying the 3D W-MSA on the deformation field patches is not enough for dealing with the stitching effect see Figure 3. Hence, based on the 3D W-MSA outputs, we further use the 3D SW-MSA to enhance information change between windows for stitching effect alleviating. We follow the cyclic-shifting strategy and move the window along the diagonal direction by one region $\left(\frac{w/n}{m} \times \frac{d/n}{m} \times \frac{h/n}{m}\right)$. By cyclic-shifting, the generated window goes from $n^3$ to $(n+1)^3$. As shown in Figure 1, after the 3D SW-MSA, the window number increased from 8 to 27. With the increase of windows, regions from different windows are mingled for calculation, which allows information exchanges to erase the stitching effect. After the two modules, we stitch the self-attended window to obtain the stitched deformation field. We underline that our stitching strategy is like a clip-on function to refine the deformation field without introducing an additional loss term. A detailed description of the architecture of W-MSA and SW-MSA can be found in (Liu et al., 2021b). The detailed algorithm description can be found in Algorithm 1.

---

**Algorithm 1:** Training Plan

---

▸ Training Process

**Input:**  unaligned image $x$,  reference image $y$,
**Output:**  deformation field $z_{y \to z}$,  registered result $x \circ \psi_{z_{y \to z}}$

**for** *each training pair* **do**

   1. Cut $x$ and $y$ into multiple patches: $p_i^x$ and $p_i^y$, where $i \in [1, 2, 3, ..., n^3]$;

   2. Feed each paired of patches $p_i^x$ and $p_i^y$ into the same-weight 3D CNN for feature extraction.

   3. For all CNN-extracted patch features pairs, concatenate them and input them into a decoder to restore the patch-wise deformation field until all patch feature pairs have been restored. .

   4. Use "3D W-MSA" and "3D SW-MSA" to stitch all the patch-wise deformation fields to get the image-level deformation field $\psi_{z_{x \to y}}$

   5. Apply the stitched deformation field $z_{x \to y}$ to the moving image $x$ via a Spatial Transformer Network (STN) to obtain the registered result $x \circ \psi_{z_{x \to y}}$.

   6. Ensure quality registration by calculating the total loss, comprising the reconstruction loss ($\mathcal{L}_{recon}$), smooth loss ($\mathcal{L}_{smooth}$), and the patchwise contrastive loss ($\mathcal{L}_{contrast}$).

**end**

---

### 3.4   Testing Scheme with Unified Registration and Segmentation.

During the testing process, we first fed the trained network with the moving image $x$ and fixed images $y$ to obtain two deformation field $\hat{z}_{y \to x}$ and $\hat{z}_{x \to y}$. It's important to note the distinct roles of these fields, $\hat{z}_{x \to y}$ is for registration, and $\hat{z}_{y \to x}$ is for segmentation: (1). *For registration*, we used a spatial transform network (Jaderberg et al., 2016) to warp the moving image $x$ and the deformation field $\hat{z}_{x \to y}$ to get the registered image $x$. (2). *For segmentation*, We then used a spatial transform network to warp the segmentation mask of fixed image $y_{seg}$ and the deformation field $\hat{z}_{y \to x}$, we then obtain the segmentation results of the moving image $(y_{seg} \circ \psi_{\hat{z}_{y \to x}})$. With a single GPU (NVIDIA A100 GPU), our method can process around 3.2 brain images per second.

## 4   Experimental Results

In this section, we detail the set of experiments performed to evaluate our proposed unified framework.

| Task | Technique | C-WM | C-CX | L-V | L-WM | L-CX | T-P | Cau | Put | - |
|---|---|---|---|---|---|---|---|---|---|---|
| Registration | *VoxelMorph* | 0.777 | 0.829 | 0.791 | 0.682 | 0.815 | 0.849 | 0.795 | 0.804 | - |
| | *SYMNet* | 0.787 | 0.840 | 0.778 | 0.717 | 0.838 | 0.883 | 0.825 | 0.853 | - |
| | *DeepTag* | 0.770 | 0.847 | 0.792 | 0.755 | 0.885 | 0.870 | 0.823 | 0.830 | - |
| | *CycleMorph* | 0.806 | 0.859 | 0.809 | 0.733 | 0.860 | 0.848 | 0.811 | 0.828 | - |
| | *ViT-V-Net* | 0.812 | 0.850 | 0.820 | 0.741 | 0.860 | 0.859 | 0.828 | 0.870 | - |
| | ***PC-SwinMorph*** | **0.825** | **0.880** | **0.847** | **0.801** | **0.908** | **0.892** | **0.859** | **0.880** | - |

| | Technique | Pal | 3-V | 4-V | B-S | Hipp | Amy | CSF | V-DC | AVG |
|---|---|---|---|---|---|---|---|---|---|---|
| | *VoxelMorph* | 0.734 | 0.644 | 0.733 | 0.881 | 0.654 | 0.659 | 0.611 | 0.790 | 0.753 |
| | *SYMNet* | 0.787 | 0.657 | 0.710 | 0.865 | 0.680 | 0.675 | 0.599 | 0.805 | 0.769 |
| | *DeepTag* | 0.753 | 0.680 | 0.725 | 0.892 | 0.710 | 0.679 | 0.545 | 0.804 | 0.773 |
| | *CycleMorph* | 0.748 | 0.663 | 0.757 | 0.891 | 0.685 | 0.652 | 0.614 | 0.791 | 0.772 |
| | *ViT-V-Net* | 0.760 | 0.669 | 0.779 | 0.897 | 0.721 | 0.664 | 0.610 | 0.793 | 0.783 |
| | ***PC-SwinMorph*** | **0.802** | **0.676** | **0.784** | **0.913** | **0.763** | **0.698** | **0.614** | **0.824** | **0.812** |

| Task | Technique | C-WM | C-CX | L-V | L-WM | L-CX | T-P | Cau | Put | - |
|---|---|---|---|---|---|---|---|---|---|---|
| Segmentation | *VoxelMorph* | 0.776 | 0.831 | 0.783 | 0.698 | 0.812 | 0.860 | 0.802 | 0.830 | - |
| | *SYMNet* | 0.729 | 0.781 | 0.767 | 0.790 | 0.769 | 0.726 | 0.780 | 0.803 | - |
| | *DeepTag* | 0.759 | 0.842 | 0.786 | 0.758 | 0.883 | 0.867 | 0.812 | 0.834 | - |
| | *CycleMorph* | 0.812 | 0.869 | 0.787 | 0.759 | 0.869 | 0.864 | 0.811 | 0.844 | - |
| | *ViT-V-Net* | 0.820 | 0.880 | 0.799 | 0.771 | 0.872 | 0.867 | 0.811 | 0.850 | - |
| | ***PC-SwinMorph*** | **0.834** | **0.885** | **0.847** | **0.806** | **0.912** | **0.888** | **0.855** | **0.878** | - |

| | Technique | Pal | 3-V | 4-V | B-S | Hipp | Amy | CSF | V-DC | AVG |
|---|---|---|---|---|---|---|---|---|---|---|
| | *VoxelMorph* | 0.764 | 0.673 | 0.738 | 0.881 | 0.663 | 0.669 | 0.574 | 0.797 | 0.759 |
| | *SYMNet* | 0.758 | 0.764 | 0.771 | 0.764 | 0.777 | 0.664 | 0.771 | 0.760 | 0.760 |
| | *DeepTag* | 0.756 | 0.659 | 0.708 | 0.887 | 0.718 | 0.696 | 0.538 | 0.800 | 0.771 |
| | *CycleMorph* | 0.776 | 0.680 | 0.737 | 0.891 | 0.709 | 0.690 | 0.570 | 0.801 | 0.780 |
| | *ViT-V-Net* | 0.777 | 0.681 | 0.781 | 0.895 | 0.755 | 0.700 | 0.584 | 0.809 | 0.791 |
| | ***PC-SwinMorph*** | **0.802** | **0.681** | **0.786** | **0.915** | **0.781** | **0.718** | **0.610** | **0.823** | **0.817** |

Table 1: Numerical comparisons between our proposed PC-SwinMorph technique and SOTA techniques on the CANDI dataset. The *Dice* similarity metric for each region is listed. We denote the regions as C-WM (L/R-Cerebral-WM), C-CX (L/R-Cerebral-CX), L-V (L/R-Lateral-Vent), L-WM (L/R-Cerebellum-WM), L-CX (L/R-Cerebellum-CX), T-P (L/R-Thalamus-Proper), Cau. (L/R-Caudate), Put. (L/R-Putamen), Pal. (L/R-Pallidum), 3-V (3rd-Vent), 4-V (4rd-Vent), CSF, B-S (Brain-Stem), Hipp. (L/R-Hippocampus), Amy. (L/R-Amygdala), V-DC (L/R-VentralDC). The average of the *Dice* metric over all regions is presented in the last column. The best performance is highlighted in bold font.

## 4.1 Datasets Description

We evaluate our framework using two publicly available datasets: the Child and Adolescent Neuro Development Initiative (CANDI) dataset(Kennedy et al., 2012) and the LONI Probabilistic Brain Atlas (LPBA40) dataset(Shattuck et al., 2008).

**CANDI Dataset.** CANDI dataset is comprised of 103 T1-weighted MRI scans with anatomic segmentation labels. The volume size of the MRI scans ranges from $256 \times 256 \times 128$ to $256 \times 256 \times 158$ voxels with a uniform space of $0.9375 \times 0.9375 \times 1.5$ $mm^3$. To prepare the segmentation mask for usages in the testing stage, we adopted the convention outlined in (Wang et al., 2020), which involves grouping corresponding organs from the left and right hemispheres of the brain. This process results in the formation of 16 distinct anatomical regions, excluding the background. We highlight that the segmentation masks are only used during the testing phase in our study. For computational efficiency, we crop the volume to $160 \times 160 \times 128$ around the centre of the brain, which is large enough to incorporate the whole brain region.

| TASK | TECHNIQUE | FRONT | PAR | OCC | TEMP | CING | PUT | HIPP | AVG |
|---|---|---|---|---|---|---|---|---|---|
| Registration | *VoxelMorph* | 0.866 | 0.709 | 0.683 | 0.797 | 0.679 | 0.648 | 0.654 | 0.720 |
| | *SYMNet* | 0.880 | 0.741 | 0.712 | 0.821 | 0.708 | 0.729 | 0.689 | 0.755 |
| | *DeepTag* | 0.893 | 0.756 | 0.745 | 0.845 | 0.736 | 0.743 | 0.710 | 0.775 |
| | *CycleMorph* | 0.889 | 0.747 | 0.735 | 0.849 | 0.723 | 0.758 | 0.705 | 0.772 |
| | *ViT-V-Net* | 0.895 | 0.756 | 0.739 | 0.851 | 0.744 | 0.779 | 0.710 | 0.781 |
| | ***PC-SwinMorph*** | **0.900** | **0.767** | **0.751** | **0.860** | **0.744** | **0.780** | **0.725** | **0.790** |
| Segmentation | *VoxelMorph* | 0.871 | 0.716 | 0.702 | 0.802 | 0.686 | 0.643 | 0.667 | 0.728 |
| | *SYMNet* | 0.888 | 0.745 | 0.737 | 0.826 | 0.716 | 0.730 | 0.687 | 0.762 |
| | *DeepTag* | 0.898 | 0.758 | 0.763 | 0.848 | 0.739 | 0.742 | 0.708 | 0.780 |
| | *CycleMorph* | 0.895 | 0.758 | 0.765 | 0.851 | 0.732 | 0.758 | 0.705 | 0.782 |
| | *ViT-V-Net* | 0.895 | 0.761 | 0.768 | 0.855 | 0.739 | 0.759 | 0.710 | 0.786 |
| | ***PC-SwinMorph*** | **0.901** | **0.766** | **0.771** | **0.860** | **0.743** | **0.773** | **0.729** | **0.794** |

Table 2: Numerical comparisons between our proposed PC-SwinMorph technique and SOTA techniques on the LPBA40 dataset. The *Dice* similarity metric for each region is listed. We denote the regions as FRONT (Frontal), PAR (Parietal), OCC (Occipital), TEMP (Temporal), CING (Cingulate), PUT(Putamen), HIPP (Hippo). While the average of the *Dice* metric over all regions is presented in the last column. The best performance is highlighted in bold font.

**LPBA40 Dataset.** LPBA40 dataset contains 40 T1-weight 3D brain volumes from 40 healthy humans. The size of 3D volumes is $181 \times 217 \times 108$ with a uniform space of $1 \times 1 \times 1 \ mm^3$. The 3D brain volume was manually segmented to identify 56 structures. Similar to the CANDI dataset, we crop the data to $160 \times 192 \times 160$ around the centre of 3D volumes to reduce the size of the volume whilst preserving all the brain regions. All the 56 structures are grouped into seven large regions in order to display the segmentation results more intuitively (Liu et al., 2020).

## 4.2 Implementation Details

**Data Pre-processing.** We normalise the volumes to have zero mean and unit variance. For the CANDI dataset, we follow the data splitting in (Wang et al., 2020) and select 20 volumes as test data, 1 volume as the reference image, and the rest as training data. For the LPBA40 dataset, we set the first volume as the reference image, the next 29 images as training images, and the last 10 images as testing images.

**Training Strategy.** During the training stage, the parameters of all convolutional layers are initialised by following the initialisation protocol of (He et al., 2015). Adam optimiser is used during training with the initial learning rate setting as $10^{-3}$. The learning rate decays by 0.1 scale every 50 epochs and terminates after 200 epochs. The batch size for both datasets is 1. All models are run on an NVIDIA A100 GPU with 80G RAM, which takes around 6 hours to train the model on the CANDI dataset and around 4 hours on the LPBA40 dataset.

**Evaluation Protocol.** To make our results comparable to other state-of-the-art methods, we use the *Dice* similarity coefficient to evaluate the segmentation and registration quality of our model, which measures the overlap between ground truth masks and predicted segmentation results. *The code with detailed description will be available with the publication.*

## 4.3 Comparison to the State-of-the-Art Techniques

We compared our technique with four recent unsupervised brain segmentation methods, including VoxelMorph (Balakrishnan et al., 2019), DeepTag (Ye et al., 2021), SYMNet (Mok & Chung, 2020), CycleMorph (Kim et al., 2021), ViT-V-Net (Chen et al., 2021b). In general, all methods are based on the fundamental architecture of VoxelMorph. For a fair comparison, all models use the same backbone architecture, VoxelMorph, which has been fine-tuned to achieve optimal performance.

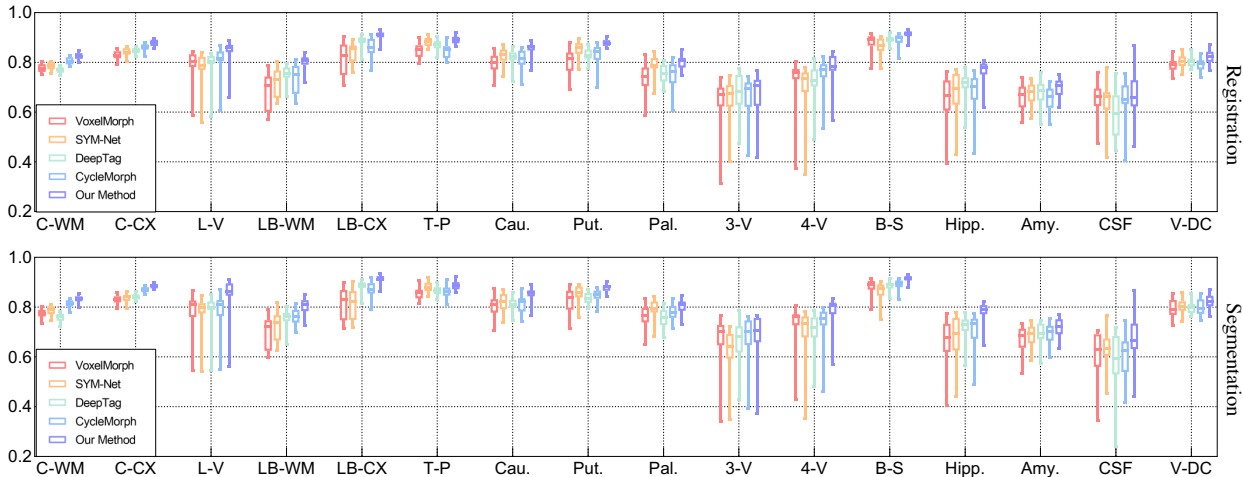

Figure 4: Boxplot comparisons between our proposed PC-SwinMorph technique and SOTA techniques on the Candi dataset. The Dice similarity metric is reported per each region in the brain. The numerics per region can be found in Table 1.

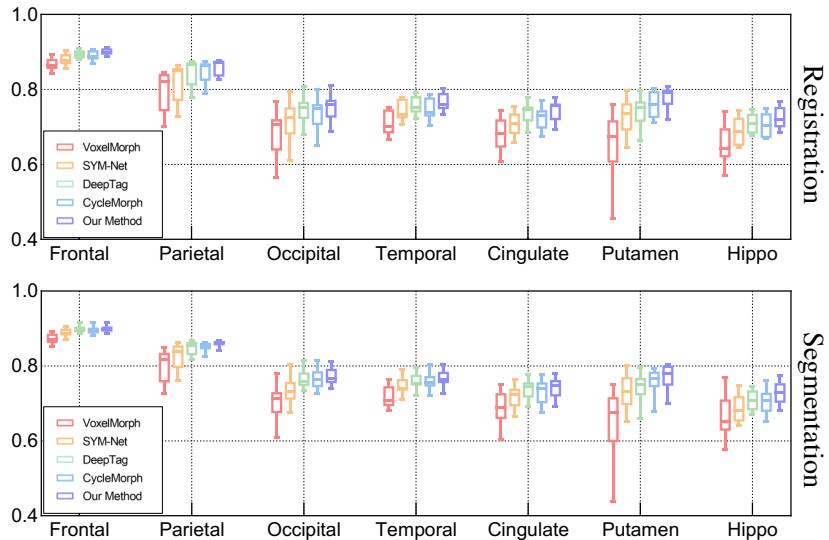

Figure 5: Boxplot comparisons between our proposed PC-SwinMorph technique and SOTA techniques on the LPBA40 dataset. The Dice similarity metric is displayed per each region in the brain. We reported the exact values of each technique in Table 2.

**Numerical Comparison.** Tables 1- 2, Fig. 4 and Fig. 5 summarise performance-wise, in terms of the *Dice* coefficient, the compared SOTA methods, and our PC-SwinMorph. The numbers are obtained using the testing scheme described in section 4.2. In a closer look at the tables, we observe that for both data, our method outperforms all other SOTA methods by a large margin, including the overall performance as well as local performance in major regions. Particularly, on the CANDI dataset our results report an improvement of 5.9% compared to VoxelMorph, and 3.9-4.3% against the other compared techniques. This performance gain is consistent on the LPBA40 dataset, where our proposed technique is 1.2-7.0% higher in performance than the SOTA methods.

Table 3: Numerical comparison of "percent of negative Jocabian determinant" and "computational time" for registration task on CANDI and LPBA40 datasets. The lower the "percent of negative Jocabian determinant" is the better the model performs. The best performance is denoted in bold font.

| Technique | LPBA40 CANDI | | | LPBA40 DATASET | | |
|---|---|---|---|---|---|---|
| | Dice | J% | Time | Dice | J% | Time |
| *VoxelMorph* | 0.753 | 5.01 | 40 | 0.720 | 5.11 | 42 |
| *SYMNet* | 0.769 | 6.34 | 49 | 0.755 | 5.97 | 50 |
| *DeepTag* | 0.773 | 4.33 | 38 | 0.775 | 4.75 | 41 |
| *CycleMorph* | 0.772 | 2.21 | 39 | 0.772 | 2.66 | 40 |
| *ViT-V-Net* | 0.783 | 2.10 | 46 | 0.781 | 2.19 | 51 |
| ***PC-SwinMorph*** | **0.812** | **1.97** | **34** | **0.790** | **2.01** | **39** |

**Smoothness of Deformation Field.** Another well-known evaluation metric is the "percent of negative Jacobian determinant values," which serves as a crucial indicator of the quality of spatial transformations in image registration tasks. A negative Jacobian determinant value signifies a local contraction or folding in the transformation, which is generally undesirable as it can indicate distortions or unnatural alterations in the image structure. Maintaining a low percentage of such negative values is indicative of a smooth, physically plausible transformation that preserves the integrity of the original image's topology. As shown in Table 3, our results showcase a lower percent of negative Jacobian determinant values. This outcome further proves that employing a multi-head shift-window mechanism to stitch image patches is a beneficial strategy.

**Computational Time.** As illustrated in Table 3, our model demonstrates a slight advantage in speed during the testing of a single pair of images. This increase in efficiency is attributed to our approach of segmenting the image into smaller patches for processing. By handling these smaller patches individually, we significantly reduce the computational burden.

**Visual Comparison.** We support the numerical results with additional visual results for our technique and the compared ones. Figure 7 shows some sample slices of the segmentation results predicted by VoxelMorph, SYMNet, DeepTag, CycleMorph, ViT-V-Net and our proposed method PC-SwinMorph. Whilst the results produced by the compared SOTA techniques are anatomically meaningful, they fail to capture fine details in several regions. By contrast, PC-SwinMorph is able not only to produce a better output but also to capture details. The zoom-in views in Figure 7 highlights these effects. Overall, PC-SwinMorph can better accommodate with fine details of the brain structure, producing segmentation closer to the ground truth. Figure 6 shows the warped images produced by different SOTA techniques as well as the proposed PC-SwinMorph. We can observe that our registration outputs are closer to the reference image, displaying fewer splitting effects from the patch generation whilst keeping fine details.

### 4.4   Ablation Study

We provide a set of experiments to further support the design of our technique.

**Contrastive Representation.** A contrastive feature learning mechanism is embedded into the registration architecture which promotes feature-level learning. The contrastive loss forces the network to contrast the difference between the two extracted CNN feature maps and therefore, the network is more discriminative to different images via contrasting unaligned images and reference images. As shown in Table 4, *the testing results demonstrate this new mechanism significantly improves the segmentation performance by around 2-3% upon the baseline model.*

**Patchwise Contrastive Learning.** Based on the idea of contrastive learning, we introduce a patchwise contrastive learning strategy. It uses a multilayer patch-based approach rather than operating on entire images. The Patchwise contrastive loss introduced in equation 6 encourages two corresponding patches, in the image, to map to a point in a learned feature space, at the same time drawing negative if they match to other patches. This mechanism further boosts the registration performance and produces better

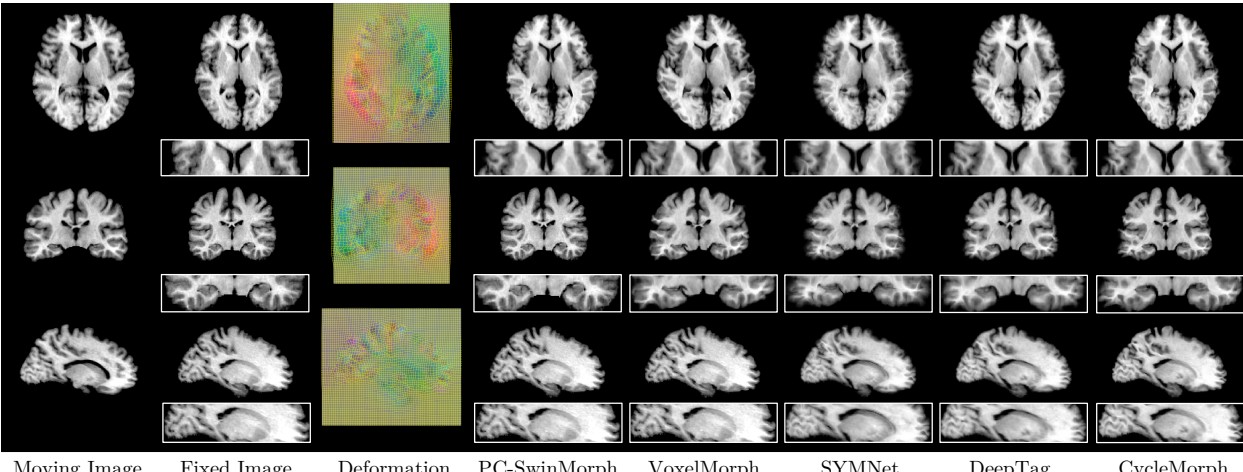

Figure 6: Visual comparisons between the proposed PC-SwinMorph technique, and other unsupervised SOTA techniques for registration. The rows show the three views from 3D volumes. The first two columns display the moving and fixed images, while columns 4-8 present the aligned images produced by PC-SwinMorph and other unsupervised SOTA techniques. The zoom-in views highlight regions that demonstrates the improvement of our method in terms of preserving the global brain structures and fine local details. The third column presents the deformation field computed by our method.

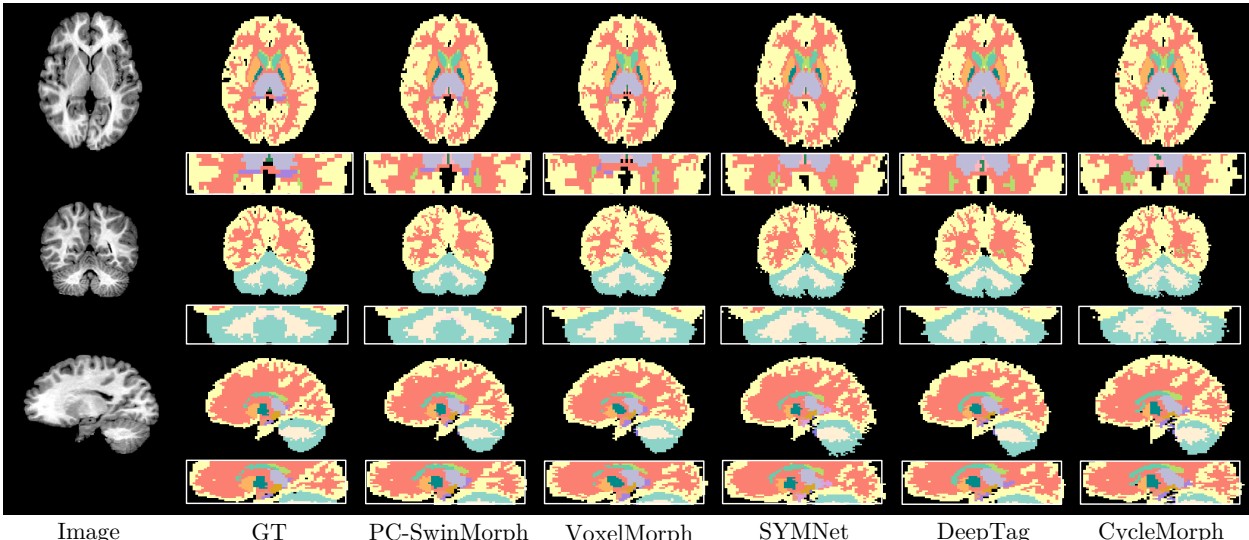

Figure 7: Visual comparisons between the proposed PC-SwinMorph technique, and other unsupervised SOTA techniques for segmentation. The rows show the three views from 3D volumes. The second column displays the ground truth (GT) results whilst columns 3-6 display predicted segmentation from PC-SwinMorph and other unsupervised SOTA techniques. Zoom-in views demonstrate that the proposed method preserved more details in different regions thus producing better segmentation results.

segmentation results. From the *Dice* coefficient comparison reported in Table 4, we can observe that the patchwise contrastive approach offers an additional 2% improvement with respect to the contrastive learning approach. This improvement is observed in both the registration and segmentation performance. This introduced strategy offers an overall improvement of 4% when compared to the baseline model.

|  | CANDI | | LPBA40 | |
|---|---|---|---|---|
|  | *Reg.* | *Seg.* | *Reg.* | *Seg.* |
| Baseline (B) | 0.753 | 0.759 | 0.720 | 0.728 |
| B + PW | 0.779 | 0.777 | 0.751 | 0.762 |
| B + CL | 0.771 | 0.772 | 0.760 | 0.766 |
| B + PW + CL | 0.791 | 0.792 | 0.779 | 0.783 |
| B + PW + CL + SW | **0.812** | **0.817** | **0.791** | **0.794** |

Table 4: Ablation study on both CANDI and LPBA40 datasets. The best performance is highlighted in bold font. We use the abbreviation 'Seg.' and 'Reg.' for Segmentation and Registration respectively. Measures are averaged over all regions using the *Dice* coefficient. We denote the baseline as 'B', patchwise method as 'PW', contrastive loss as 'CL', and window/shifted-window multi-head self-attention method as "SW".

**The effect of 3D W-MSA & 3D SW-MSA.** One of the main drawbacks of patchwise learning is the lack of information exchange between patches. The introduction of 3D W-MSA and 3D SW-MSA, which stitches the patches and enhances the performance across patches, has been demonstrated to be greatly effective. As shown in Table 4, the segmentation and registration performance on the CANDI dataset has been improved by 2.65% and 3.06% respectively when using 3D SW-MSA. This performance behavior is prevalent on LPBA40 demonstrating consistent performance.

## 5 Discussion

From the experiments, we observe several strengths on our model. Firstly, our hypothesis that patch embeddings are a more meaningful representation is supported by our experiments. We observed that our technique has a significant (statistically) performance than the current SOTA techniques. *What is the intuition behind our technique?* The spatial structure of the patch that allows capturing not only global but, more importantly, also local anatomical representations. These local and global representations are reflected in capturing fine-grained details and, therefore, helping the registration to be more robust to the changes in the to-be registered images. Secondly, we highlight that our model is not a trivial combination between VoxelMorph and SwinTransformer. Literature on ViT and VoxelMorph uses the off-the-shelf ViT for better feature extraction, i.e., directly replacing the CNN encoder of the VoxelMorph with ViT. However, our motivation for using SW MHA is to stitch deformation field patches. We underline that we did not change the original CNN encoder part of VoxelMorph with SW MHA. Instead, we only use the SW MHA (one layer from SwinTransformer) to stitch patches after the encoder/decoder part of VoxelMorph. Because we only use one layer of SwinTransformer, the computational cost with a negligible increase – the Flops of the model without SW MHA is 416.04 GFlops, whilst for our model are 416.10 GFlops, which only increases 0.6 GFlops.

## 6 Conclusion

We introduce a novel unified unsupervised framework for image registration and segmentation. We propose to rethink these tasks from a patch-based perspective and introduce two patch-based strategies. Firstly, we introduce a novel patch-based contrastive strategy to obtain richer features and preserve anatomical details. Secondly, we design a new patch stitching strategy to eliminate any inherent artifact from the patch-based partition. Our intuition behind the performance gain of our strategies is that through patches we capture not only global but also local spatial structures (more meaningful embeddings). We demonstrated that our technique reported SOTA performance for both tasks.

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
