# OpenReview forum: "PC-SwinMorph: Patch Representation for Unsupervised Medical Image Registration"
_TMLR — Rejected by TMLR_

### Review · Reviewer_84PG · 2023-12-02

**Summary Of Contributions:**

This submission suggests an unsupervised formulation for medical image registration where multi-head self-attention modules in the image processing model are used. Another main claimed novel aspect is the adoption of contrastive loss. Authors investigate the combination of patchwise registration with self-attention modules to achieve patchwise alignment and introduction of contrastive loss for each pixel. The image objects involved in this work are of 3D.

**Audience:**

Yes

**Claims And Evidence:**

Yes

**Requested Changes:**

(1) Authors should state the mathematical formulation of the _image registration problem_ clearly in the script as well as the main proposed method.

(2) Is any of the following evaluation metrics for the registration quality applicable to the submitted work?

    a) mean and variance of the squared sum of intensity differences
    b)the cross-correlation loss itself
    c) Normalized RMSE
    d) voxel-wise two-norm of the deformation error
    e) the Hausdorff distance
    f) the Local Correlation Coefficient (LCC) metric
    g) harmonic energy
    h) relative variance log-Jacobian

(3) Runtime comparison between mentioned methods in the manuscript can be helpful to understand the time complexity of the proposed approach.

(4) Difference with precedent work: [ViT-V-Net: Vision Transformer for Unsupervised Volumetric Medical Image Registration, Chen et al.] also considers using transformers for image registration problem. Is this method a proper baseline to benchmark with?

(5) In section 4.1, authors state “_...forming overall 16 anatomical regions_”. Does this imply 16 different copies of networks are trained with respect to each anatomical region respectively? If there is only one model trained, authors can share insights on the generalization properties of the proposed algorithm.

(6) In section 4.2 Testing Scheme paragraph, authors mentioned the procedure where segmentation is followed by a wrapping operation. Should there be description in section 3 about how segmentation is involved in the registration problem?
I suggest authors give a clear mathematical description in section 3 about the training and testing setup respectively.

(7) Polishing the writing: I suggest authors thoroughly parse the manuscript and adjust English expressions from place to place.
For instance, in the paragraph 3D W-MSA on page 7, ‘_Whereas the W_MSA calculates region relations inside windows, which can largely save computation time._’; in the first paragraph on page 8 “_Whilst the detailed description of ... can be found in ...._”.These are not correct (complete) English sentences.

(8) Mathematical notation: $\circ$ is usually used to denote function composition. With the warping operation by a spatial transformer denoted as $\mathcal{W}$, by the expression $x\circ \psi_{z_{x\rightarrow y}}$, do authors mean $\mathcal{W}(x, \psi_{z_{x\rightarrow y}} )$? Notations in [A Deep Learning Framework for Unsupervised Affine and Deformable Image Registration, de Vos et al.] appear clearer to me.

(9) Shifting window: on page 7, paragraph 3D SW-MSA, I would like to understand the details of the “cyclic-shifting strategy”. Are patches/tokens shifted and fed into the self-attention layers? Please give further explanation/description.

(10) To further improve the clarity, it can be helpful if authors can state, in the format of an algorithm, the exact steps to setup the training method proposed in this manuscript.

**Strengths And Weaknesses:**

Strength:
The narrative of the manuscript is clear and the structure is well set up.

Weakness:

(1) Important technical details are either not revealed in full or discussed with ambiguity, in particular mathematical definitions of key operations proposed in this work. Specific questions are stated in the following section.

(2) Evaluation metrics reported in this work only concern Dice similarity metric. Is this metric the sole criterion to evalaute the image registration quality?

---

### Review · Reviewer_g67j · 2023-12-05

**Summary Of Contributions:**

The paper seeks to register medical volumes (3D images) by a series of steps:

Divide the two volumes into box-shaped patches.
Feed the patches into parallel CNNs to get semantic features.
Use a decoder (in a way I did not understand) to generate a deformation map to register one volume on the other.
Stitch the deformed volume x (hopefully close to volume y).
Use a Transformer to smooth the stitched volume.

**Audience:**

Yes

**Claims And Evidence:**

No

**Requested Changes:**

Abstract
The first sentence is vague. More generally, this is not really a medical paper in the sense of addressing a concrete medical use case. While registration is important in some cases (medical, more so in neuroscience imaging), the use cases mentioned in the Introduction are not realistic. For example, would any surgeon use unsupervised registration to prepare for brain surgery? Also, the Results section has no evaluation of whether the registration would serve any particular medical use case. Perhaps the Introduction can be reworded to reflect this general goal of improving unsupervised registration, or to cite use cases where the proposed method would provide sufficiently accurate registrations and segmentations.

Last sentence page 1:
"Lack of high quality prior knowledge": (a) Is there a reference for this? (b) This sounds like the exact thing the proposed method is choosing to not use. Does the method effectively substitute for this missing prior knowledge?

The literature review looks thin (pgs 1 and 2). Is this all the work there is in this field?

Page 2:
"Medical images are more complex..." (a) This is not obvious to me. (b) I did not see an explanation later of how the proposed method accounts for the fine detail. The encoder CNNs, for example, extract high level semantic features, which appears to contradict the need stated in this sentence.

2.1:
Are there good review papers that could be cited?

"unify the registration and segmentation": Can you give details as to why this might be advantageous? It seems like once you have the registration, transfer of segmentations is straightforward.

2.2:
If Liu is the closest work, is there more to be said about how the proposed work relates to Liu?

3.1:
"wrap" -> "warp" (I think)

page 5:
Line 4:
The indexing of the patches is confusing. i is in 1 to n, but there are n^3 patches.

"two-symmetric weight-shared 3D encoder": Is this standard terminology? If not, can it be explained with more detail?

"highly semantic": It sounds like this would by definition miss all low-level features such as edges and textures.

"Their corresponding tow deformation field patches": It sounds like the rectangular patches are the atomic elements of the deformation map, and that they retain their rectangular shape. How does the map warp them?

L_{ncc}: what is the effect when pixel magnitudes change, eg bright vs dark images? It seems this would derail the NCC loss.

Patchwise Contrastive Loss: provide reference.

Page 6:
Top 3 lines: (a) There is a huge imbalance between positives and negatives: n^3 to n^6 I think. How is this handled? (b) I don't understand why x_i and y_i are paired as positives. They do not correspond to matching parts of the unregistered (unmapped) images. (c) (Important) This patch pairing looks like supervised learning. Evidently I am missing some part of the description.

Eqn 4:
If tau = 1, why include it?

3.3:
Can you give motivation for using a Transformer to smooth the stitched image? (a) Is it better than other smoothing methods? (b) It seems that Transformers would be ill-suited for this because they lack a "local correlation" inductive bias, which smoothing (I imagine) needs.

tables 1 -  2, figs 3 - 4:
These should, I think, be moved to the Results section. Their current location is confusing.

Tables 1, 2:
(Important) Can you include +/- std devs on the accuracies? This is important in order to assess whether the differences between methods are in a noise envelope.

Fig 5, 6:
Perhaps there is a zoomed example that would be more illustrative. I do not see any differences in the examples shown.

Discussion:
"The spatial ... fine-grained details": As noted earlier, I don't see how fine-grained detail is targeted, given the high-level semantics generated by the encoders which then inform the deformation.

**Strengths And Weaknesses:**

I think I may not be an ideal reviewer for this paper. I did not understand the description of the algorithm. Maybe it requires more familiarity with the structures and jargon than I have.

My main concern is that the description of the algorithm is perhaps not yet optimal. See specific notes below, which itemize spots where a well-intentioned reader can lose the thread.

My entry under "Audience" is a total guess.

Going forward, I feel it would be unproductive for me to review further versions of the paper. For this reason none of my comments require changes.

---------------------------------------------

Note to TMLR editors:

1. Please include line numbers in the latex template. They make reviewing so much easier.

2. The bibliography style makes finding references pointlessly difficult. Please edit the bibliography style so that last names, as used in the text body, are easy to find in the bibliography. Examples: Use numbers; use initials for first names; put the surname first.

---

### Review · Reviewer_Fn8D · 2023-12-22

**Summary Of Contributions:**

The authors introduce a novel framework employing a patch-based approach for unsupervised medical image registration and segmentation. It proposes unique strategies like patchwise contrastive strategy and patch stitching using 3D window/shifted-window multi-head self-attention modules. These methods enhance feature representation and detail alignment. The framework demonstrates improved performance on two benchmark datasets.

**Audience:**

Yes

**Broader Impact Concerns:**

The authors claim that the key contribution is the two patch-based strategies, in which patchwise contrastive loss is a key part of the unsupervised medical image registration task. The authors tried to explain this process with some figures I assume, but I cannot find where they are in the manuscript. I guess any reader will have some feelings if the authors do not address this issue.

**Claims And Evidence:**

Yes

**Requested Changes:**

We want the authors:
- Please try to solve the weakness mentioned above.
- Please carefully proofread the submitted draft; for now, it looks like it is not a complete draft.

**Strengths And Weaknesses:**

Strengths
- The paper introduces a somewhat novel patch-based framework for unsupervised medical image registration and segmentation.
- The patchwise contrastive strategy and patch stitching with 3D window/shifted-window multi-head self-attention modules improve detail alignment and feature representation.
- The framework shows improved performance on major medical benchmark datasets (CANDI and LPBA40), indicating its effectiveness and reliability.

Weakness
- It is very confusing what is the detailed patchwise contrastive mechanism, and in which figure?  This is very important for the readers to understand the key innovation.

---

### Decision · Action_Editor_D1MT · 2024-03-29

**Recommendation:** Reject

**Comment:**

There are multiple claims that are not supported with best scientific practice.  Statistical significance is asserted, but apparently not actually tested - the numbers seem different enough that there should likely be significance in many cases, but it is far from evident that it will hold in all places, especially if a Bonferroni correction is needed.  Sweeping generalizations that patch based embeddings are a "more meaningful" representation do not appear to be sufficiently well defined to be supported by experiments.

The submission would need a major revision to address these issues, as the main issues are in scientific methodology and not just editing unsupported claims.

**Audience:**

The submission is appropriate for a TMLR audience, addressing a machine learning approach to medical image registration.

**Claims And Evidence:**

The submission makes claims that (i) they introduce a novel patch based machine learning method for image registration, (ii) a novel patch stitching method, (iii) "demonstrate from the numerical and visual results that... two patch-based strategies lead to better
performance than the state-of-the-art techniques for unsupervised registration".  Additional claims are made in the text that (iv) "Medical images are more complex than natural images", (v) "patch embeddings are a more meaningful representation", (vi) "PC-SwinMorph then enforces more meaningful feature representation", (vii) "in this work, we use this observation to unify the registration and segmentation process within one framework, as the unification can be advantageous as it allows for simultaneous refinement of both tasks, potentially leading to more accurate and cohesive results."

There were some disagreements among the reviewers about whether the TMLR criteria were fulfilled, but some significant concerns remain about whether evidence is sufficiently provided for the above claims.  Some of them could be addressed by a minor revision, such as removing claims that "medical images are more complex than natural images" without appropriate definitions or quantitative evidence of "complexity".  Other issues are more central to the content of the paper, such as the lack of stated statistical testing.  "Significance" or "statistically" is claimed several places, but no stated hypothesis tests, significance levels, p-values, or confidence intervals are given.  Thus, an appropriate scientific methodology is not present in the submission and numerical performance claims are not appropriately supported.

Detailed comments from the action editor:

Last paragraph before bullet point contributions in Section 1 - claims that are not quantifiable/supported from the data in the paper: "Medical images are more complex than
natural images", "patch embeddings are a more meaningful representation", "PC-SwinMorph then enforces more meaningful feature representation"

End of 2.1 - is segmentation a claimed contribution?  If so, should it be a bullet point in Section 1, if not, don't slip in an extra claim

Figure 3 caption English mistake: "yields results with clear discontinues between patches"

Equation (6) states that it is a weighted sum, but does not seem to have weighting terms

Tables 1,2,3,4 place highest number in bold without a stated significance test / p-value.

Bottom of page 13 etc. states percent improvements without confidence intervals.

Discussion: "our hypothesis that patch embeddings are a more meaningful representation is supported by our experiment" - it is not clear what "meaningful" means and the discussion does not sufficiently tie this claim back to quantitative experiments.

"We observed that our technique has a significant (statistically) performance than the current SOTA techniques" - The test should be specified and p-values / confidence intervals should be provided to make this statement.

Discussion of flops should be contextualized by discussing size of images and relationship with standard medical imaging practice.

Quoting from the reviewer discussion:

(1) The writing quality of this submission still needs overhaul after one round of revision. Specific improvements that are still needed include attaining alignment in mathematical notations with other published work such as [A Deep Learning Framework for Unsupervised Affine and Deformable Image Registration, de Vos et al., Medical Image Analysis 2019], [CycleMorph: Cycle Consistent Unsupervised Deformable Image Registration, Kim et al., Medical Image Analysis 2021], [Adversarial Uni- and Multi-modal Stream Networks for Multimodal Image Registration, Xu et al., MICCAI 2020] and [Medical image registration, Hill et al., Physics in Medicine & Biology 2000].

(2) Now that authors formulate the problem with a reconstruction loss and still state in their reply that most reconstruction metrics did not apply for quality evaluation are contradictory. [Deep Learning in Medical Image Registration: A Survey, Haskins et al., 2020] lists criteria such as sum of squared differences (SSD), cross-correlation (CC), mutual information (MI) [72, 106], normalized cross correlation (NCC), and normalized mutual information (NMI). Unfortunately, authors fail to report relevant numerical metrics to illustrate the advantages of the proposed methods.

(3) Most importantly, the main novel point in this submission stated in section 3.3 lack sufficient clarity. Authors fail to explain clearly what inputs are to attention layers and and what the shifting technique means exactly. Also, the details of the function family adopted to embody the transformation fields are not stated. The ambiguity in the statement of the main novelty of the submission is a major concern.

My conclusion is that this submission should be firmly rejected due to the lack of clarify in its methodology, insufficiency of numerical evidence, and the poor writing quality.

**Resubmission Of Major Revision:**

The authors may consider submitting a major revision at a later time.